# Adapting the Opening Minds Stigma Scale for Healthcare Providers to Measure Opioid-Related Stigma

**DOI:** 10.3390/pharmacy12040105

**Published:** 2024-07-09

**Authors:** Ashley Cid, Alec Patten, Michael A. Beazely, Kelly Grindrod

**Affiliations:** School of Pharmacy, University of Waterloo, 10A Victoria St. S, Kitchener, ON N2G 1C5, Canada; s2patten@uwaterloo.ca (A.P.); kelly.grindrod@uwaterloo.ca (K.G.)

**Keywords:** opioid crisis, opioid use disorder, naloxone, community pharmacy, stigma

## Abstract

The opioid crisis in Canada continues to cause a devastating number of deaths. Community-based naloxone programs have been identified as one of the solutions for combatting this crisis; however, there are disparities in which pharmacies stock and offer naloxone. Opioid-related stigma is a major barrier for limited naloxone distribution through pharmacies. Therefore, the development of anti-stigma interventions is crucial to improve naloxone distribution in Canada. However, there is no validated tool to specifically measure opioid-related stigma. The Opening Minds Stigma Scale for Healthcare Providers (OMS-HC) is a validated scale used to measure mental illness-related stigma. This study will adapt the OMS-HC by using four different opioid-related terminologies to determine which is the most stigmatizing to use in an opioid-related anti-stigma intervention. Pharmacy students completed four versions of the adapted OMS-HC. The average OMS-HC scores and Cronbach’s α co-efficient were calculated for each version. The term “opioid addiction” was found to be the most stigmatizing term among participants and will be used in the adapted version of the OMS-HC in a future anti-stigma interventions.

## 1. Introduction

The Canadian opioid crisis continues to see an escalation in opioid-related toxicity deaths [1]. A total of 40,642 deaths have occurred in Canada between January 2016 and June 2023 [1]. Of the accidental apparent opioid toxicity deaths that have occurred so far in 2023, most (84%) have involved fentanyl and opioids that are non-pharmaceutical (80%) [1]. Naloxone, a temporary antidote for an opioid overdose, can be administered to those experiencing opioid-induced respiratory depression (OIRD). In 2016, Health Canada removed naloxone from the prescription drug list, allowing naloxone to be dispensed by a community pharmacist without a prescription [2]. Pharmacists are one the most accessible frontline healthcare providers, which puts them in an ideal position to play a role in harm reduction by providing education and naloxone to two groups: (1) patients and their friends or family who may be at risk of OIRD or (2) members of the general public who could be in a position to help someone who is experiencing OIRD [3]. For example, most Ontarians (91%) live within five kilometers of a pharmacy and many community pharmacies are open extended hours and, in some instances, open 24 h, which increases patient accessibility and convenience [4,5]. It is not mandatory for community pharmacies in Canada to participate in the naloxone distribution program, leading to a high degree of variability in the number of pharmacists who stock and/or dispense naloxone [6]. For example, in 2018, only half (55.6%) of Ontario pharmacies dispensed naloxone, and one third (33.7%) of kits dispensed were from the top 1.0% of naloxone dispensing pharmacies [7]. There is a demonstrated need for increased and equitable naloxone distribution through community pharmacies to further promote harm reduction; however, barriers that prevent proactive naloxone dispensing need to also be addressed.

Pharmacists frequently interact with patients who consume regulated and unregulated opioids, including patients with opioid use disorder (OUD). The care that they provide to this population includes dispensing their medications, referring to other care providers, identifying high risk use, providing medication education, recommending and monitoring medications for OUD, promoting medication adherence, and offering/dispensing naloxone [8,9]. The literature shows that healthcare professionals, including pharmacists, are prone to stigmatizing patients with substance use disorders, which results in suboptimal care provided for these patients [10]. In fact, stigma is described as a major barrier preventing pharmacists from proactively offering naloxone kits to patients at risk of OIRD [11]. Werremeyer et al. conducted a survey study where they examined the degree in which pharmacists prefer to maintain a greater social distance from patients with opioid misuse and opioid use disorder using a social distance scale (SDS) [12]. The SDS assesses a person’s willingness to interact with a target person in different types of relationships, showcasing their attitude toward the individual [12]. Werremeyer et al. discovered that of the 187 pharmacists who completed the survey, the mean SDS score was 16.32 (range 9–23), where higher scores represented a greater preference for social distance [12]. Over half of the pharmacists (59%) had a SDS score greater than 15, which demonstrated an overall lack of willingness to interact and stigmatization towards patients who consume opioids or who have an OUD [12]. Werremeyer et al. conducted another study where they implemented a training program for pharmacists to reduce stigma towards people who misuse opioids and measured social distance scores through a survey pre- and post-program [13]. The program resulted in significantly lower social distance scores immediately post-program when compared to the baseline score (14.75 vs. 16.57, *p* = 0.017) [13].

The Mental Health Commission of Canada published a report on stigma and the opioid crisis in 2019 [14]. One key recommendation for combatting opioid-related stigma was for organizations to develop comprehensive stigma reduction and intervention strategies for front-line providers [14]. The report also highlighted that robust evaluation and monitoring frameworks should be incorporated in interventions to ensure programs meet the objective of reducing stigma [14]. Policymakers, professionals, and organizational leaders should also lead initiatives for increasing the use of non-stigmatizing language and establish best practice guidelines for opioid-related terminology and language [14]. A scoping review of what is known about community pharmacy-based take-home naloxone programs identified that educational programs aimed at reducing stigma for pharmacists should be implemented to promote proactive naloxone dispensing and increased distribution [11]. In order to be successful, studies must be able to systematically and accurately measure stigmatizing attitudes among pharmacists.

In response to the need to address stigma in community pharmacies and to optimize naloxone dispensing, the authors developed the Optimizing Naloxone Dispensing in Pharmacies (ONDP) online continuing education program with a full protocol previously published [15]. The goals of the ONDP program were to increase the amount of naloxone kits dispensed by increasing knowledge about naloxone, increasing confidence and motivation to proactively offer naloxone, and decreasing stigma associated with naloxone and opioids [15]. Currently, there is a lack of validated stigma scales to measure opioid-related stigma in healthcare professionals, and the evidence base supporting anti-stigma interventions related to opioid use is thin, with very little Canadian-based research [16]. The authors selected the previously validated “Opening Minds Stigma Scale for Healthcare Providers” (OMS-HC) to be able to measure stigma related to opioids at various timepoints during the ONDP program study. For example, participants of the ONDP program will complete a pre-program survey which will include the 15-item OMS-HC, then participants will complete a post-program survey including the same 15-item OMS-HC. Participants will then be invited to implement what they learned from the program in their pharmacies, and after 3 months have passed, they will be invited to complete a follow-up survey which will include the same 15-item OMS-HC. The authors will use these three timepoints to identify any changes in the level of stigma reported by the participants and to assess if the program in itself has decreased stigma. The ONDP study will be run as a randomized controlled trial whereby the control group will also complete the 15-item OMS-HC at the same three timepoints to assess for any changes in the level of opioid-related stigma.

The OMS-HC was developed in Canada within the framework of the “Opening Minds” anti-stigma initiative [17]. The scale was developed using focus groups consisting of healthcare providers and people with lived experience of mental illness and was tested with 787 healthcare providers, including pharmacists, across Canada [18]. The 20-item scale showed good internal consistency, with a Cronbach’s α of 0.82 and satisfactory test–retest reliability [18]. The OMS-HC was only weakly correlated with social desirability, indicating that social desirability was not likely to be a major factor in OMS-HC scores [18]. Social desirability bias is a type of response bias that occurs when participants in a research study submit responses based on what they believe will make them appear better to others, concealing their true opinions or experiences [19]. A follow-up study determined that, although the internal consistency of the 20-item scale was satisfactory, factor analysis revealed that five items could be potentially removed, suggesting a fifteen-item version of the scale is superior to the full twenty-item scale [20]. Therefore, the authors decided to use the 15-item scale in this study. Each question in the scale uses a 5-point Likert scale to quantify stigma [20]. Higher scores suggest a more stigmatizing attitude [20]. The OMS-HC has further demonstrated good construct validity and internal consistencies (α = 0.67–0.79) [20]. The scale measures three subtypes of stigma: attitudes, disclosure, and social distance [20]. The attitudes’ subscale measures how healthcare professionals view people who have a mental illness [21]. These views can include seeing people with a mental illness as dangerous or viewing people with a mental illness with compassion [21]. The disclosure subscale focuses on a healthcare professionals’ willingness to tell close friends or colleagues that they themselves have a mental illness [21]. The social distance subscale measures a healthcare professionals’ willingness to be in close physical proximity, such as working close together or living close to a person who has a managed mental illness [21].

The purpose of this study is to adapt the 15-item OMS-HC to measure stigma related to opioid use in pharmacy professionals. An adaptation of the OMS-HC for opioid-related stigma has not been previously studied. There are multiple ways to refer to someone who consumes opioids or who has an opioid substance use disorder; therefore, it is important to understand which terminology is the most stigmatizing to use in the adapted OMS-HC: opioid use disorder, opioid addiction, opioid dependency, and/or opioid misuse/opioid use disorder. The authors chose to look for the most stigmatizing term to allow for the most sensitive and accurate measure of opioid-related stigma in the ONDP study. We hope that our adaptation of the OMS-HC questionnaire to measure stigma will provide an important tool for researchers to employ in the context of interventions to reduce stigma associated with opioid use.

## 2. Materials and Methods

### 2.1. Study Design

The OMS-HC was adapted into four different survey versions to reflect four terminologies commonly used for opioid-related substance use. Based on the previous experience and expertise of M.B. and A.C., we decided to test 4 versions of the survey: version 1 used the terminology “opioid dependency”, version 2 used the terminology “opioid use disorder”, version 3 used the terminology “opioid addiction”, and version 4 used the terminology “opioid misuse/use disorder” (all 4 versions are available in the Appendix A). The wording of the questionnaire was kept the same as the original 15-item OMS-HC, except the wording of “mental illness” was directly replaced with either one of the four different terminologies. Each question had a 5-point Likert scale rating with scores ranging from 1 (strongly disagree) to 5 (strongly agree). The total scores range from 15 to 75, with lower scores indicating a less stigmatizing attitude. In addition to the OMS-HC scale, participants were also asked to rank the four terminologies from least stigmatizing (1) to most stigmatizing (4), where the version that had the highest sum would be considered the most stigmatizing. Therefore, the survey contained a total of 16 questions.

### 2.2. Study Setting and Population

The survey was hosted online using Qualtrics to collect participant data. The study obtained ethics approval from the University of Waterloo Research Ethics Board (REB #43796, Approved 15 September 2023). The pilot study population included a cohort of 119 third-year pharmacy students from the University of Waterloo School of Pharmacy who were enrolled in a pharmacy therapeutics course coordinated by M.B. This cohort was selected due to opioid use disorder being a part of the curriculum in which they were studying at the time and that they would have the most pharmacy-related work experience out of all the cohorts due to having completed three co-operative education terms and would be considered ready to enter the pharmacy workforce. The pharmacy cohort was randomly divided into four groups of approximately 30 students using a random number generator by A.P. Each participant received an e-mail with a study recruitment letter and a survey link. After consent to participate was obtained, each student completed one of the four survey versions that was assigned. Upon completing the survey, each student received a 0.25% bonus participation grade in the pharmacy therapeutics course. If students did not consent to participate in the survey, they had the opportunity to instead write a short reflection statement on opioid use disorder and stigma and submit it to A.C. as an alternative measure for obtaining the 0.25% bonus participation grade. Students were given three weeks to complete their assigned survey, which closed on 17 December 2021.

### 2.3. Data and Statistical Analysis

No demographic parameters were collected due to all participants being recruited from a common source. Participant responses to the survey were downloaded into an Excel spreadsheet (Microsoft Corp., Redmond, WA, USA). Data analysis included calculating the average, standard deviation, and the minimum and maximum scores for each version of the OMS-HC. The sum of each of the four opioid terminologies for the stigma ranking question was also calculated. Reverse coding was applied for the questions that required it for each version of the OMS-HC. The internal consistency for each version of the OMS-HC was evaluated using the Cronbach’s α coefficient, similar to the studies conducted by Kassam et al. and Modgill et al., which use the OMS-HC in their analyses [18,20]. The Cronbach’s α co-efficient is a commonly used measure of internal consistency for a test or scale, or in other words, how closely related a set of items are as a group or the extent in which all the items in a test measure the same concept [22]. It is expressed as a number between 0 and 1 [22]. Internal consistency is important to be determined before a test or scale is used for research purposes to ensure validity [22]. A general accepted rule is that a Cronbach’s α of 0.6–0.7 indicates an acceptable level of reliability, and 0.8 or greater is considered to be very good [23].

## 3. Results

Of the 119 pharmacy students in the cohort, 91 (77%) completed and submitted their respective version of the OMS-HC survey. Version 1 of the survey initially included 29 participants, while the remaining versions each initially included 30 participants. Of the completed surveys, 22 participants completed version 1 (76%), 20 participants completed version 2 (67%), 24 participants completed version 3 (80%), and 23 participants completed version 4 (77%). Table 1 displays the minimum and maximum scores for each survey version along with the corresponding average OMS-HC score, the standard deviation, the Cronbach’s α, and the totals for the stigma rankings. Version 2 (opioid use disorder) had the highest OMS-HC average score of 38.70, while version 3 (opioid addiction) had the second highest score of 37.33. Version 1 (opioid dependency) had the lowest score of 35.54 with version 4 (opioid misuse/use disorder) having a very similar score of 35.74. Versions 2–4 had similar standard deviations: 6.67, 6.72, and 7.23, respectively. By contrast, version 1 had a slightly higher standard deviation of 9.96. The internal consistencies for each of the versions ranged from 0.63 to 0.9. When participants were asked to rank the four terminologies from least stigmatizing (1) to most stigmatizing (4), version 3 (opioid addiction) was ranked the most stigmatizing with a summative score of 294 while version 2 (opioid use disorder) was ranked the least stigmatizing with a summative score of 196. Versions 4 (opioid misuse/use disorder) and 1 (opioid dependency) had very similar summative scores of 202 and 208, respectively.

## 4. Discussion

The average OMS-HC scores were similar amongst the four survey versions. Version 2 (opioid use disorder) and version 3 (opioid addiction) both had higher average OMS-HC scores, 38.70 and 37.33, respectively, indicating a higher level of stigma when these terms are used. Generally, the term “opioid use disorder” is considered a less stigmatizing term amongst researchers and healthcare professionals as it is considered direct medical terminology that describes problematic opioid use as an illness rather than a term of individual blame [24]. It is not surprising to the authors that the term “opioid addiction” had a high OMS-HC average score as this wording is considered to be highly stigmatizing language [24]. Version 1 (opioid dependency) and version 4 (opioid misuse/use disorder) had the lowest OMS-HC average scores of 35.54 and 35.74, respectively. When comparing these scores to the initial study conducted by Modgill et al., which validated the 15-item OMS-HC scale, the “allied health” group which consisted of pharmacists/pharmacy students had an average OMS-HC score of 33.3 and a standard deviation of 6.7, which is similar to the results demonstrated in our study [20]. Sherwood conducted a study looking at healthcare curriculum influences on stigma towards mental illness [25]. A group of pharmacy, nursing, and social work students were surveyed using a 12-item OMS-HC scale (minimum score 12, maximum score 60) before and after taking a psychiatric course [25]. The pharmacy students’ pre-course mean OMS-HC score was 30.9 and post score was 32.24, demonstrating a more negative stigma among students after the course [25]. Sherwood discusses that didactic courses likely do not reduce stigma towards mental illness, and generally if a course were to require students to reflect on their thoughts and viewpoints on patients’ conditions then there is a higher likelihood for stigma to decrease [25]. The average scores between the study conducted by Sherwood and this one are similar when considering the maximum score achievable on each respective scale [25]. Of note, the pharmacy students who completed the course in the study by Sherwood were second-year students, in comparison to third-year pharmacy students who completed this study; however, the therapeutics course completed by the third-year pharmacy students was also considered to be a didactic course; therefore, the ONDP program should incorporate self-reflection activities to increase the likelihood that the intervention will decrease opioid-related stigma [25].

The Cronbach’s α for our study ranged from 0.63 to 0.9, while Modgill et al. reported a similar internal consistency with an overall Cronbach’s α of 0.79 and a range of α = 0.67–0.68 in the three subscales (attitude, disclosure, and social distance), which is generally considered acceptable [20]. The study conducted by Sherwood involving the measurement of stigma before and after pharmacy students took a psychiatry course demonstrated a Cronbach’s α of 0.78 [25]. Another study conducted by Van der Maas et al. examined the application of the OMS-HC survey in the community health centre setting [26]. They aimed to assess the level of stigma held by community health centre staff towards clients with mental and/or substance use problems in the Greater Toronto Area (Canada) [26]. The OMS-HC was found to also have a similar acceptable internal reliability for the 15-item version of the scale (α = 0.77) and a range of α = 0.67–0.79 for its subscales [26]. The authors of this study discussed that the OMS-HC was shown to correlate with a series of scales commonly used in stigma research (i.e., The Mental Illness Clinician’s Attitudes Scale, the Marlowe–Crowne Social Desirability Scale, the Recovery Assessment Scale, the Recovery Self-Assessment Provider Version, and the Bogardus Social Distance Scale); therefore, the OMS-HC scale alone was found to be appropriate for use and advantageous over the use of multiple scales [26].

When examining the stigma ranking scores for the four versions, version 3 (opioid addiction) had the highest ranked score for the most stigmatizing term. This does not come as a surprise as the word “addiction” has increasingly been considered to be a stigmatizing term in the literature [24]. For example, a study conducted by Goodyear et al., looked at the role of gender, language, and precipitating events and their impact on stigma related to opioid use [27]. They conducted a randomized, between-subjects case vignette (n = 2605) using an online survey [27]. Participants rated a hypothetical individual with an opioid use disorder on different dimensions of stigma after seeing one version of a vignette that varied by three conditions: (1) a male versus a female, (2) an individual labelled as being a “drug addict” versus having an “opioid use disorder”, and (3) an individual whose use started by taking prescription opioids from a friend versus prescribed by a physician [27]. The findings showed that there were higher stigmatizing attitudes overall towards a male, an individual labeled as a “drug addict” and an individual who took prescription opioids from a friend [27]. For people with a substance use disorder, stigma stems from the inaccurate belief or myth that “addiction” is a moral weakness, instead of what it is known to be—a chronic, treatable disease from which patients can recover and lead a healthy life [24]. Therefore, version 3 (opioid addiction) will be implemented in the ONDP program due to its high OMS-HC average score and high stigma ranking.

In 2022, the Mental Health Commission of Canada developed a scale to measure opioid-related stigma, called: “The Opening Minds Provider Attitudes Towards Opioid-Use Scale (OM-PATOS) [28]. Knaak et al., who are the same group of authors that developed the OMS-HC, completed an exploratory and confirmatory factor analysis of the OM-PATOS [28]. Exploratory factor analysis findings demonstrated a 15-item 2-factor solution, with subscales of “attitudes” (6 items) and “behaviours/motivation to help” (9 items) using the terminology “opioid use problem” [28]. The confirmatory factor analysis provided preliminary confirmation of the factor structure but further research with a larger sample size is needed [28]. Overall, the results of their factor analysis supported the use of the 15-item scale with health professionals and first responders but the authors noted that further studies are needed to validate the study to the same level as the OMS-HC [28]. Therefore, we will use the adapted version of the OMS-HC using the wording of “opioid addiction” in the ONDP program to measure stigma changes in pharmacy professionals over time.

This study has several limitations, including the small sample size, which makes it difficult to compare the consistency of the OMS-HC across four different survey versions and the fact that the study population used included pharmacy students and not pharmacists. The ONDP program will recruit licensed pharmacists and pharmacy technicians; therefore, if this study had recruited pharmacists and pharmacy technicians instead of students, then there is a chance that the results would be more similar to what would be expected in the results of the ONDP program. Additionally, any self-report measure, including the original iteration of the OMS-HC for mental illness, will be subject to self-report or social desirability bias [29]. Social desirability bias may operate in self-report measures and cannot be excluded from the OMS-HC or any other stigma scale based on self-report [30,31]. Another potential limitation is that, even though stigma scales like the OMS-HC offer a reasonable indication of the beliefs that support behavioural attitudes, they have limitations in capturing actual behavioural responses [20]. Future evaluations of anti-stigma interventions targeting healthcare providers should measure the impact and health outcomes of patients to determine the downwards effect of the interventions [20]. Additionally, future research on anti-stigma interventions should incorporate additional measures for external validity, such as those that measure knowledge and intended behaviour [18].

## 5. Conclusions

Substance-use stigma has made it difficult to reframe the opioid crisis as a public health issue and has acted as a barrier for patients looking to access evidence-based treatments and other lifesaving medications like naloxone. There are multiple studies in the area of mental health-related stigma reduction which are relevant and useful, but have not been previously applied to the field of opioid-related stigma. Therefore, intervention research is needed to establish the effectiveness of stigma reduction strategies in the opioid substance use field. This study aimed to determine which out of four opioid-related substance use terminologies would best be suited in an adapted version of the OMS-HC. It was determined based off the average OMS-HC score and a self-reported ranking of the terminologies among participants that the term “opioid addiction” was the most stigmatizing and should be used in the adapted version of the OMS-HC. This adapted version of the OMS-HC will be used in an anti-stigma intervention targeted towards increasing naloxone distribution in pharmacists and pharmacy technicians where stigma will be measured pre-, post-, and after three months. This intervention will also measure knowledge, confidence, and motivation to proactively offer/dispense naloxone to patients which will give a good measure of external validity. The establishment of a multi-component intervention such as the ONDP program is crucial to improve the level of care provided to patients, increase access to treatment for opioid use disorder, and increase naloxone distribution across Canada.

## Figures and Tables

**Table 1 pharmacy-12-00105-t001:** Data results for each of the four versions of the OMS-HC, including the summative scores of the stigma ranking.

Study Version	OMS-HC Minimum	OMS-HC Maximum	Average	Standard Deviation	Cronbach’s α	Stigma Ranking (Total)
1, Opioid dependency	22	63	35.5	10.0	0.90	208
2, Opioid use Disorder	27	53	38.7	6.7	0.79	196
3, Opioid addiction	22	56	37.3	6.7	0.63	294
4, Opioid misuse/use disorder	22	46	35.7	7.3	0.75	202

## Data Availability

The original contributions presented in the study are included in the article/Appendix A, further inquiries can be directed to the corresponding author/s.

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
