# Peer review of "Adapting the Opening Minds Stigma Scale for Healthcare Providers to Measure Opioid-Related Stigma"

_pharmacy, 2024, doi:10.3390/pharmacy12040105_

Round 1

Reviewer 1 Report

Comments and Suggestions for Authors

I read with interest the paper titled "Adapting the Opening Minds Stigma Scale for Healthcare Providers to Measure Opioid-related Stigma"

- The first part of the paper is written with pharmacists at the center of the question, instead highlighting the role of pharmacies. My first question is if Pharmacy Technicians also take a role in the community pharmacies in those kind of activities (i believe yes, since in the last part, Pharmacy Techncians are refered as well). From a quick search, it seems that the role of community pharmacist does not end up in the pharmacists. So please discuss that and give the reader more details.  

- The overall introduction seems too lenghtly and repetitive. Please reduce the detail in the scales. The citation is there, so no need to explain this in such detail. 

- Briefly explain the meaning of stigma ranking - it better to be high or low? From the conclusions it seems to be good to be high, which is quite surprisingly. 

""opioid addiction” was the most stigmatizing and should be used in the adapted version of the OMS-HC" - is that correct? Why should the most stimatizing term be used, instead of the least? Please explain it clearly. 

Author Response

Thank you very much for your insightful comments and suggestions!

Comment 1: The first part of the paper is written with pharmacists at the center of the question, instead highlighting the role of pharmacies. My first question is if Pharmacy Technicians also take a role in the community pharmacies in those kind of activities (i believe yes, since in the last part, Pharmacy Techncians are refered as well). From a quick search, it seems that the role of community pharmacist does not end up in the pharmacists. So please discuss that and give the reader more details.  

Response 1: In Canada, only pharmacists can train patients to use naloxone. However, pharmacy technicians can play an important role in proactively offering naloxone at prescription intake, stocking kits, etc. Because we were only testing the scales in pharmacy students we haven’t discussed technicians or other pharmacy staff in the current manuscripts. The subsequent study in which we employed this scale included technicians but they were less than 5% of the respondent.

Comment 2: The overall introduction seems too lenghtly and repetitive. Please reduce the detail in the scales. The citation is there, so no need to explain this in such detail. 

Comment 2: We have reduced unnecessary content throughout including in the introduction (e.g. lines ~106, ~110, ~141, etc. as well as in the discussion) You can review these deletions on the resubmitted version with tracked changes.

Comment 3: Briefly explain the meaning of stigma ranking - it better to be high or low? From the conclusions it seems to be good to be high, which is quite surprisingly. 

Response 3: We have ensured that the meaning of the stigma ranking is clear:

“Higher scores suggest a more stigmatizing attitude.20 ” - line 119 in the introduction, also, “The total scores range from 15 to 75, with lower scores indicating a less stigmatizing attitude ” line 150 in the methods section

Comment 4: ”"opioid addiction” was the most stigmatizing and should be used in the adapted version of the OMS-HC" - is that correct? Why should the most stimatizing term be used, instead of the least? Please explain it clearly. 

Response 4: The authors chose to look for the most stigmatizing term to allow for the most sensitive, and accurate, measure of opioid-related stigma in the ONDP study. We have clarified this in the introduction (line 135). The rationale for choosing the most stigmatizing language is that it should give us higher baseline scores and should be able to more easily detect reductions in stigma after an intervention.

Reviewer 2 Report

Comments and Suggestions for Authors

Dear authors,

Many thanks for submitting the manuscript.

The paper reads well an the english used is good.

The paper is tackling an delicate issue but one commonly experienced and a major barrier for patients coming forward and it is highlighting the need to have this addressed by the population.

The use of 4 groups and comparison between the 4 different terms using a tool already validated in Mental Health research is beneficial.

There are a couple of comments

Table 1 could benefit from all results on a flat line rather than some higher than others

Line 350 there is a quotation mark but not required

On the whole a good clear paper with good methodology and comparisons.

Having the questionnaire as an appendix may also be of interest to the readers

Best wishes

Author Response

We appreciate your comments!

Comment 1: Table 1 could benefit from all results on a flat line rather than some higher than others

Response 1: We have replace table 1 with one that is better-formatted

Comment 2: Line 350 there is a quotation mark but not required

Response 2: Quotation mark has been deleted

Comment 3: Having the questionnaire as an appendix may also be of interest to the readers

Response 3: We have added all 4 versions as a supplementary file

Reviewer 3 Report

Comments and Suggestions for Authors

The authors raise an important problem in opioid use disorder that escalates from year to year. Any research that allow to understand and raises the awareness about the treatment of opioid addiction is very valuable and should be promoted.

However, there are some issues that should be explained.

1. Is the naloxone kit free? If not - is the price the same in every pharmacy?

2. Whether all pharmacists in Ontario are trained how to educate people who want to use the naloxone kit?

3. Which form of the drug exactly does naloxone kit contain? Are they safe to use? 

4. Are there any data available showing demand of naloxone kit depending on the location of the pharmacy (e.g. in the city or countryside or depending on the city district). 

5. What is the correlation between unequal supply of pharmacies in naloxone and the stigmatization of opioid addiction.

6. The expected impact of this study on stigma reduction should be clearly stated.

Author Response

Really interesting questions and comments!

Comment 1: Is the naloxone kit free? If not - is the price the same in every pharmacy?

Response 1: Naloxone kits are free from any pharmacy in Canada. However, there are multiple naloxone dosage forms available - intramuscular injections and intranasal devices. Only two provinces, Ontario and Quebec, distribute both forms free of charge via pharmacies. In the Northwest Territories, only the intranasal form is available. In all other provinces, only the injectable form is available free of charge (So R, Al Hamarneh Y, Barnes M, Beazely MA, Boivin M, Laroche J, Patel H, Sihota A, Smith T, Tsuyuki RT. The status of naloxone in community pharmacies across Canada. Can Pharm J (Ott). 2020 Sep 21;153(6):352-356).

Comment 2: Whether all pharmacists in Ontario are trained how to educate people who want to use the naloxone kit?

Response 2: Pharmacists are not required to be trained prior to dispensing naloxone. That knowledge would be the responsibility of the pharmacists and they would treat it as any new drug on the market. However, many pharmacists participated in naloxone training programs once it was removed from the prescription drug list in 2016 (however we don’t have any references to support this). Naloxone is taught at all pharmacy schools in Canada (I don’t have a reference for this either - other than discussions with colleagues).

Response 3: Which form of the drug exactly does naloxone kit contain? Are they safe to use? 

Response 3: Naloxone kits typically contain two intramuscular naloxone injections or two intranasal naloxone dosage forms. Naloxone is very safe and does not typically cause significant adverse effects if administered inappropriately (e.g. to someone in medical distress but not due to an opioid overdose)

Comment 4: Are there any data available showing demand of naloxone kit depending on the location of the pharmacy (e.g. in the city or countryside or depending on the city district). 

Response 4: To my knowledge - no. We do have data on # of kits dispensed from pharmacies (as well as via public health units, other service providers) but we wouldn’t have data about the # of individuals who would like a kit but do not have access at their local pharmacies.

Comment 5: What is the correlation between unequal supply of pharmacies in naloxone and the stigmatization of opioid addiction.

Response 5: We can only speculate. As mentioned in the Choremis study (reference 7) naloxone distribution via pharmacies was very unequal, particularly early on. Factors relating to this unequal distribution likely include how many individuals are requesting naloxone, whether or not the pharmacy is part of a chain or group where naloxone distribution is encouraged, local needs, the amount of opioid agonist therapy patients receiving care at the pharmacy, other local access to naloxone (e.g. via a public health unit), as well as stigma.

Comment 6: The expected impact of this study on stigma reduction should be clearly stated.

Response 6: The intention of modifying the scale was not to reduce stigma reduction per se, rather, it is to modify the OMS-HC scale for opioid use so that we can employ the scale in other studies (including one of our own) that has an intervention to reduced stigma.

We have added at the end of the introduction: “We hope that our adaptation of the OMS-HC questionnaire to measure stigma will be provide an important tool for researchers to employ in the context of interventions to reduce stigma associated with opioid use. 

Reviewer 4 Report

Comments and Suggestions for Authors

 - The authors should add the date of receiving the Institutional Review Board Statement;

 - The Introduction section is too long, I recommend shortening it.

 - Some of the information in the Introduction section should be provided in the Discussion section.

 - The study's aim is mentioned twice (Lines 132 - 133 and Lines 140 - 142).  Be more precise and clearly define the purpose of the study.

 - Lines 134 - 146 - This information is better moved to the Materials and Methods section.

 - I recommend using the following subsections in the “Material and methods” part: (a) study design (b) study setting; (c) study population; (d) inclusion and exclusion criteria; (e) study tool; and (f) statistical analysis;

 - The different versions of OMS-HC scale should be provided as a supplementary file.

 - Why only pharmacy students were enrolled in the study? Since your study's aim was to adapt OMS-HC scale for healthcare providers, it should be validated among different healthcare professionals.

 - As you mentioned in the Limitations section, the sample size is too small, also the big limitation is that the questionnaire is testen only among phramacu students.

 - The study could be considered a pilot, but in this case the title should be changed.

Author Response

Thank you for your detailed review and suggestions!

Comment 1: The authors should add the date of receiving the Institutional Review Board Statement

Response 1: Added

Comment 2: The Introduction section is too long, I recommend shortening it. Some of the information in the Introduction section should be provided in the Discussion section.

We have reduced unnecessary content throughout including in the introduction (e.g. lines ~106, ~110, ~141, etc. as well as the discussion). You can review these deletions on the resubmitted version with tracked changes.

Comment 3: The study's aim is mentioned twice (Lines 132 - 133 and Lines 140 - 142).  Be more precise and clearly define the purpose of the study.

Response 3: We have deleted the second mention

Comment 4: Lines 134 - 146 - This information is better moved to the Materials and Methods section.

Response 4: Some of this section has been moved to the beginning of the methods section

Comment 5: I recommend using the following subsections in the “Material and methods” part: (a) study design (b) study setting; (c) study population; (d) inclusion and exclusion criteria; (e) study tool; and (f) statistical analysis;

Response 5: We have added subsections to the methods with similar headings to the ones you suggested

Comment 6: The different versions of OMS-HC scale should be provided as a supplementary file.

Response 6: We have added all 4 versions as a supplementary file

Comment 7: Why only pharmacy students were enrolled in the study? Since your study's aim was to adapt OMS-HC scale for healthcare providers, it should be validated among different healthcare professionals.

Response 7: It would have been ideal to conduct this study in practising pharmacists and/or other health professionals. We conducted this study in 3rd year pharmacy students out of convenience and because we weren’t measuring stigma per se, but rather comparing the stigma scales one to the next. The scale that was most stigmatizing, using the term “opioid addiction” was then employed in our “Optimizing Naloxone Dispensing in Pharmacies” ONDP educational program - manuscript in preparation.

Comment 8: As you mentioned in the Limitations section, the sample size is too small, also the big limitation is that the questionnaire is testen only among phramacu students.

Response 8: Correct - these are the two most significant limitations.

Comment 9: The study could be considered a pilot, but in this case the title should be changed.

Response 9: We have haven’t change the title as we felt that the term “adaptation” implies that we are repurposing a questionnaire for the first time. Rather, we have described the study has a pilot in the methods section (line 157). Happy to add “pilot” to the title if you feel strongly about it!

Round 2

Reviewer 4 Report

Comments and Suggestions for Authors

The manuscript is suitable for publication after the revision made by the authors.